# Boundary Denoising for Video Activity Localization

**Mengmeng Xu**[1][*]   **Mattia Soldan**[1][*]    **Jialin Gao**[2]    **Shuming Liu**[1]
**Juan-Manuel Pérez-Rúa**[3]    **Bernard Ghanem**[1]

[1]King Abdullah University of Science and Technology (KAUST)
[2]National University of Singapore    [3]Meta AI

## Abstract

Video activity localization aims at understanding the semantic content in long, untrimmed videos and retrieving actions of interest. The retrieved action with its start and end locations can be used for highlight generation, temporal action detection, etc. Unfortunately, learning the exact boundary location of activities is highly challenging because temporal activities are continuous in time, and there are often no clear-cut transitions between actions. Moreover, the definition of the start and end of events is subjective, which may confuse the model. To alleviate the boundary ambiguity, we propose to study the video activity localization problem from a denoising perspective. Specifically, we propose an encoder-decoder model named DenoiseLoc. During training, a set of temporal spans is randomly generated from the ground truth with a controlled noise scale. Then, we attempt to reverse this process by boundary denoising, allowing the localizer to predict activities with precise boundaries and resulting in faster convergence speed. Experiments show that DenoiseLoc advances several video activity understanding tasks. For example, we observe a gain of +12.36% average mAP on the QV-Highlights dataset. Moreover, DenoiseLoc achieves state-of-the-art performance on the MAD dataset but with much fewer predictions than others.

## 1 Introduction

The summation of human experience is being expanded at a prodigious rate, making it imperative to devise efficient and effective information retrieval systems to aid the need for knowledge abstraction and dissemination. Recently, video data has emerged as the largest unstructured knowledge repository (Kay et al., 2017; Ran et al., 2018), making it essential to develop algorithms that can understand and identify semantically relevant information within videos (Wu et al., 2019; 2017). Our research focuses on the video activity localization domain (Gao et al., 2017; Zhao et al., 2017), which enables users to identify, classify, and retrieve interesting video moments. Video activity localization tasks are defined to predict a set of temporal spans relevant to either a fixed class taxonomy (Caba Heilbron et al., 2015) or free-form natural language queries (He et al., 2019). These algorithms have numerous applications, including action segmentation (Wang et al., 2020), highlight generation (Lei et al., 2021a), product placement (Jiao et al., 2021), and video editing (Neimark et al., 2021).

Technical solutions for activity localization often draw inspiration from innovations in object detection, which is a well-studied problem in the image domain. An analogy can, in fact, be drawn between spatially localizing objects and temporally localizing moments, with the key difference being that temporal boundaries are subjective and open to interpretation (Wang et al., 2020; Alwassel et al., 2018). A prevailing approach for model design in object detection is to adopt an encoder-decoder design paired with a suitable training protocol (Carion et al., 2020; Lei et al., 2021b). For video activity localization, the encoder processes the raw video frames and, optionally, a language query, with the goal of generating rich semantic representations that capture the interaction within the video and between video and language. These representations are referred to as "memory". The decoder leverages the encoder's memory to produce a list of temporal locations with corresponding

---

[*]Denotes equal contribution.

confidence scores. The decoder achieves this by inputting candidate spans, which can be predefined based on ground truth activity locations statistics or learned during training.

The primary challenge in video localization tasks stems from the ***boundary ambiguity*** mentioned earlier. Unlike object boundaries, activities are continuous in time, and the saliency of a temporal event changes smoothly due to its non-zero momentum. Thus, transitions between activities are not always clear or intuitive. Moreover, human perception of action boundaries is instinctive and subjective. This phenomenon is reflected in the existing video datasets, where we can identify multiple clues indicating that the variance of localization information is higher than for object locations. To support this thesis, DETAD (Alwassel et al., 2018) conducted a campaign to re-annotate the ActivityNet (Caba Heilbron et al., 2015) dataset to estimate the annotator's agreement and quantify the severity of boundary uncertainty in humans. Additionally, a recent study (Lei et al., 2021a) found that over 10% of the queries in QV-Highlights showed disagreement on the boundary location with an Intersection over Union (IoU) of 0.9. Thus, addressing the uncertainty of action boundaries is crucial for developing reliable and accurate video localization pipelines.

In this study, we aim to address the challenge of uncertain action boundaries in video activity localization. To this end, we propose DenoiseLoc, an encoder-decoder model which introduces a novel **boundary-denoising training** paradigm. In DenoiseLoc, the transformer-based encoder captures the relations within and across modalities. The decoder is provided with learnable proposals and noisy ground truth spans and progressively refines them across multiple decoder layers. In detail, we iteratively extract location-sensitive features and use them to update the proposal embeddings and spans in each decoder layer. Our boundary-denoising training jitters action proposals and serves as an augmentation to guide the model on predicting meaningful boundaries under the uncertainty of initial noisy spans. Our denoising training method is easily generalizable to several video domains (i.e., YouTube videos, movies, egocentric videos) as it does not require hand-crafted proposal designs, and the inference process is the same as in a generic encoder-decoder design (Lei et al., 2021b). Surprisingly, we find that with very few proposals (*i.e.*, 30 per video), our model retrieves sufficient high-quality actions and performs on par with or better than computationally expensive methods evaluating thousands of proposals.

To demonstrate our model's effectiveness and generalization ability, we conducted extensive experiments on different datasets.

In Fig. 1, we probe our intermediate outputs on the *val* split of QV-Hightlights (Lei et al., 2021a). The model gradually denoises the boundary to the ground-truth location through multiple layers and reaches 45.29% Average mAP, without a dedicated learning scheme. On the more recent and challenging MAD dataset (Soldan et al., 2022) for activity localization in long-form videos, we achieve significant improvements of 2.69% on Recall@1.

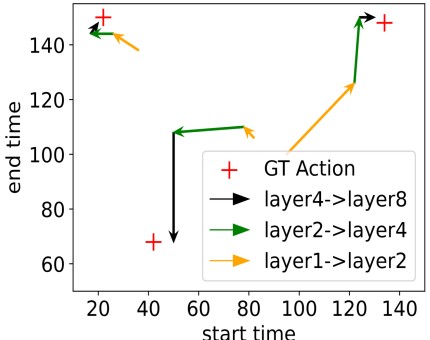

Our contributions are: **(1)** A novel boundary-denoising training approach for video activity localization tasks. The noise injected at training time leads to faster convergence with a much smaller number of queries. **(2)** DenoiseLoc, an encoder-decoder style multi-modality localization model, specifically designed to exploit the boundary-denoising training strategy. **(3)** Extensive experiments to demonstrate the effectiveness of DenoiseLoc, achieving state-of-the-art performance on several datasets, including the large-scale MAD dataset. Additionally, we provide thorough ablation and analysis studies to investigate the design choices and modeling properties of our approach.

**Figure 1: Illustration of the boundary denoising process.** Arrows show the predicted noise from each of our decoder layers, visualized on the `start-end` boundary map. The Average mAP of `layer1`, `layer2`, `layer4`, and `layer8` outputs are increasing as 38.48%, 42.07%, 44.57%, and 45.29%, evaluated on the *val* split of QV-Hightlights (Lei et al., 2021a).

## 2 RELATED WORK

**Video Activity Localization.** Video Activity Localization is the task of predicting the temporal span in a video corresponding to a human-generated natural language description. Many solutions have been proposed for this task; however, methods can be clustered into two groups: **(i)** proposal-based methods (Jiyang et al., 2017; Anne Hendricks et al., 2017; Soldan et al., 2021; Songyang et al., 2020; Escorcia et al., 2019; Xia et al., 2022), which produce confidence or alignment scores for a predefined set of $M$ temporal moments and **(ii)** proposal-free (Lei et al., 2021b; Zeng et al., 2020; Mun et al., 2020; Shaoxiang & Yu-Gang, 2020; Rodriguez et al., 2020; Li et al., 2021), which directly regress the temporal interval boundaries for a given video-query pair. Proposal-based methods tend to outperform regression-based methods, but the latter have much faster inference times since they don't require exhaustive matching to a set of proposals. Our proposed DenoiseLoc follows the proposal-free pipeline, and it is evaluated our model on both short-form (QV-Highlights (Lei et al., 2021a)) and long-form (MAD (Soldan et al., 2022)) grounding datasets.

**Denoising and Diffusion.** The issue of boundary noise has been extensively studied in the field of weakly supervised temporal action localization, where temporal annotation information is unavailable. To mitigate noise in generated pseudo labels, Yang, *et al.* (Yang et al., 2021; 2022) designed an uncertainty-aware learning module UGCT, while Li et al.(Li et al., 2022b) proposed a novel Denoised Cross-video Contrastive (DCC) algorithm to reduce the negative impacts of noisy contrastive features. In fully supervised localization, Huang *et al.* (Huang et al., 2022) introduces Elastic Moment Bounding (EMB) to accommodate flexible and adaptive activity temporal boundaries with tolerance to underlying temporal uncertainties. However, these studies mainly focus on noise modeling rather than denoising. Particularly, EMB predicts the action location and uncertainty and proposed a new evaluation metric to measure the range of predicted boundaries. In contrast, our method directly predicts the denoised span and follows the standard evaluation.

More recently, studies on detection have explored learning with noise. DN-DETR (Li et al., 2022a) discovers that including noisy object proposals can accelerate the training of transformer-based detectors such as DETR (Carion et al., 2020). DINO (Zhang et al., 2022) proposes to add noise to ground truth at different levels to generate positive and negative samples for training. Meanwhile, DiffusionDet (Chen et al., 2022), DiffTAD (Nag et al., 2023), and DiffusionVMR (Zhao et al., 2023) propose scheduled iterative denoising method to train a model to refine the input proposals with different noise levels. Although our work is inspired by, or similar to these methods, we show diffusion process is not evidentally better than a single denoising step, as the detection output is in a way lower dimension manifold than generated images.

## 3 METHOD

### 3.1 PROBLEM FORMULATION

We aim to localize temporal instances that are relevant to a pre-defined activity list or a natural language query. The input video is modeled as a sequence of $n_v$ snippets, $V=\{v_i\}_1^{n_v}$, each comprising $\mathcal{F}$ consecutive frames. If available, the language query is tokenized in $n_l$ elements $L=\{l_i\}_1^{n_l}$. Both inputs are mapped to feature vectors using pre-trained models and are identified as $X_v \in \mathcal{R}^{c_v \times n_v}$, $X_l \in \mathcal{R}^{c_l \times n_l}$, respectively, where $c_v$ and $c_l$ are the feature dimension of snippets and tokens.

To evaluate our model, we compare its predictions $\hat{\Psi}$ with human annotations $\Psi$. The model predicts $M$ possible instances, denoted as $\hat{\Psi}=\{(\hat{\psi}^n, \hat{c}^n)\}_{n=1}^M$, sorted by their predicted relevance. Each instance $\hat{\psi}^n=(\hat{t}_s^n, \hat{t}_e^n)$ represents the beginning $(\hat{t}_s^n)$ and ending $(\hat{t}_e^n)$ times of the $n$-th relevant instance, and $\hat{c}^n$ is its corresponding confidence score. In this work, we regard coupled temporal endpoints $(t_s, t_e)$ as a "temporal span".

### 3.2 DENOISELOC ARCHITECTURE

As illustrated in Fig. 2, our model pipeline includes three main designs: encoder, decoder, and denoising training. First, we feed the video features $X_v$ into the encoder as a list of computation blocks. The query embeddings $X_l$ are concatenated with the video feature after a projection layer when it is available. We choose multi-head attention as our basic block unit due to its ability for

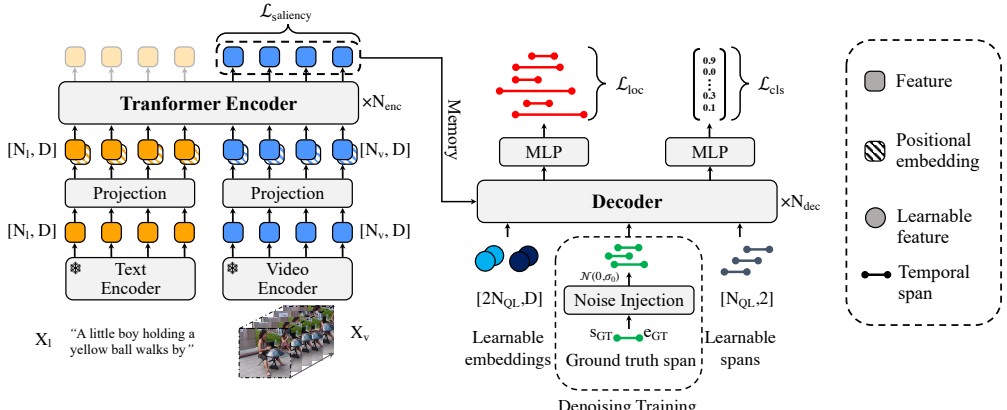

**Figure 2: Architecture.** Our proposed model consists of three main components: Encoder, Decoder, and Denoising Training. The input video and text are transformed into video and text embeddings by means of pretrained networks, then concatenated and fed to the Encoder module (left). The output features from the Encoder, termed as memory, are subsequently routed to the Decoder. During training, the Decoder receives in input a set of learnable embeddings, ground truth spans with additional artificial noise, and a set of learnable spans. The objective of the decoder is to enrich instance-level features computed based on the two sets of spans and refine (via denoising) the provided ground truth spans. Note that the GT spans are not used during inference.

long-range dependency modeling. Then, the output of the encoder (*i.e.*, memory) is forwarded to a decoder, along with a set of learnable spans. The decoder is a stack of units that apply instance feature extraction and instance-instance interaction, where we show that the boundary information can be directly used to obtain effective proposal features. The boundary-denoising training process includes a new set of noisy spans to train the decoder, and it aims to help the decoder converge faster and better. We disable those *extra spans* during inference as the ground truth spans are not available anymore. Also, we do not need NMS because a set loss was applied in training.

### 3.2.1 ENCODER

Our encoder aims at modeling semantic information of the inputs, such as multi-modality interaction and non-local temporal dependency. It includes the feature projection layer(s) followed by $N_{enc}$ transformer encoder layers. Given the video snippet features $X_v$, and optionally the language token features $X_l$, the feature project layer transforms its input into a joint space. Then, we concatenate all the available features added with the sinusoidal positional embeddings to the video features and send them to the transformer encoder layers. Since the self-attention mechanism has the potential to have message-passing over any two tokens, we assume the encoder output has rich semantic information over temporal relations and is conditioned by the given text prompt for video activity localization. Following the convention in DETR, we name this intermediate output as *memory*.

### 3.2.2 DECODER

We design $N_{dec}$ denoising layers in the decoder. Each of them works as a basic unit to predict the noise of the input spans from the video feature. Specifically, each denoising layer takes the proposal start/end and corresponding proposal embeddings as inputs and progressively refines the proposal boundaries, and classifies their categories. Such $N_q$ proposals serve as *queries* and interact with global video representation (*memory*) in the decoder, which is introduced in DETR. To avoid tedious hand-crafted designs such as pre-defined anchors, the proposal start/end and proposal embeddings are set as learnable parameters and can be updated through backpropagation.

As illustrated in Fig. 3, in each decoder layer, the module follows the order of the self-attention layer, cross-attention layer, and feed-forward layer. To be specific, the self-attention layer is adopted on the proposal embeddings to model the proposal relations with each other. Next, the cross-attention layer simulates the interactions between the proposal embeddings with the encoder's output memory, thereby the proposal embeddings can capture the rich semantic information from the video and refine the proposal boundaries. Different from DETR, we find that explicitly modeling proposal features is crucial for proposal embedding interaction. Specifically, we replace the standard cross-

attention layer in the transformer decoder with the DynamicConv (He et al., 2017) layer, which is proposed in (Sun et al., 2021). This choice is ablated and validated in Sec. 4.5, where we in investigate different architectural designs. As illustrated in Fig. 3, temporal RoI alignment (Align1D) (Xu et al., 2020) is first adopted to extract the proposal features based on the start/end locations, then the proposal features and proposal embeddings are sent into the DynamicConv module to accomplish the interactions between implicit proposal embeddings with explicit proposal features. The DynamicConv layer outputs the enhanced proposal features and the updated proposal embeddings. Subsequently, MLPs are used to predict the updated proposal start/end locations.

After each decoder layer, the proposal boundaries will be refined with predicted start/end offset. Together with the updated proposal embeddings, the refined proposals will be sent into the next decoder layer to regress more precise boundaries and more accurate classification probabilities.

### 3.2.3 BOUNDARY-DENOISING TRAINING

The boundary-denoising training module includes a new set of noisy spans to train the decoder. Therefore, the input of the decoder is the union of the learnable spans and the new noisy spans. Differently, the new noisy spans in boundary-denoising training are randomly sampled around the ground truth activity location. These sampled spans are able to diversify the temporal inputs and accelerate model convergence. However, since the activity location is unknown in the test set, we only use learnable spans during model inference.

To be more specific, during training, given a set of ground-truth temporal activities, $\{(t_s^n, t_e^n)\}_{n=1}^{M_{gt}}$ and a noise level $\sigma$, we first sample $N_{qn}$ 2D vectors from Gaussian $\mathcal{N}(0, \sigma I)$, denoted as $\{\vec{\epsilon}_n\}_{n=1}^{N_{qn}}$, $\vec{\epsilon}_n \in \mathcal{R}^2$. Then, we divide the $N_{qn}$ noise vectors into two different groups. The first group of $M_{gt}$ combines with ground truth temporal spans to create a positive set of proposals, e.g., one noisy proposal could be $(t_s^n + \epsilon_1^n, t_e^n + \epsilon_2^n), n \in \{1, \cdots, M_{gt}\}$. The second group including $N_{qn} - M_{gt}$ elements works as a negative set of proposals so that we

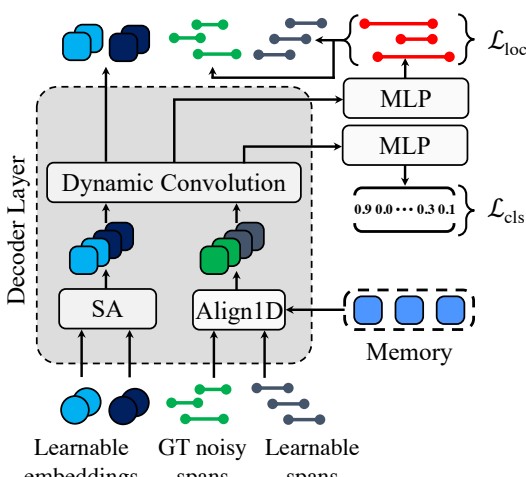

**Figure 3: Decoder.** In each decoder layer, the self-attention is first adopted on the proposal embeddings to model the proposal relations with each other. Then the proposal features are explicitly extracted by RoI alignment, and they further interact with proposal embeddings to enrich the proposal representation through Dynamic Convolution. Last, the feed-forward MLPs update proposal embeddings and start/end gradually.

directly use $(t_s^{neg} + \epsilon_1^n, t_e^{neg} + \epsilon_2^n), n \in \{M_{gt} + 1, \cdots, N_{qn}\}$ to make most of the proposals are valid (i.e., inside of the video). Note that when there is a boundary outside of $(0, 1)$ for a proposal, we will clamp it into the valid range. We have $t_s^{neg} = 0.25, t_e^{neg} = 0.75$ so that the average center and width of those proposals are half of the video duration.

Once the noisy proposals are created, we feed them into the decoder to reduce the noise in each decoder layer. Note that we still have the learned proposal embedding and spans in the decoder, but the noisy proposal does not communicate with the learned proposal in case of annotation leakage. Moreover, the noisy proposals have their own assignment to the training target, and will not affect the matching algorithm applied in the learnable spans in the decoder. In inference, our predictions are from the learned proposal position, and the decoder has *obtained* the capability to denoise the learned proposal in each layer.

### 3.3 LOSS FUNCTIONS

We denote $\hat{\Psi} = \{(\hat{\psi}^n, \hat{c}^n)\}_{n=1}^{M}$ as the set of predictions, and $\Psi = \{\psi^n\}_{n=1}^{M_{gt}}$ as the set of ground-truth temporal activities, where $\hat{\psi}^n = (\hat{t}_s^n, \hat{t}_e^n)$ and $\psi^n = (t_s^n, t_e^n)$. To compute the loss, we need to first find an assignment $\pi$ between predictions and ground truth. We determine the assignment based on the

matching cost $\mathcal{L}_{match}$ similar to DETR (Carion et al., 2020) as:

$$\mathcal{L}_{match}(\hat{\Psi}, \Psi) = \sum_{n=1}^{M_{gt}} [-\hat{c}^{\pi(n)} + \mathcal{L}_{span}(\hat{\psi}^{\pi(n)}, \psi^n)]. \tag{1}$$

With this matching cost, following previous works, we use the Hungarian algorithm (Kuhn, 1955) to find the optimal bipartite matching $\hat{\pi}$, where $\hat{\pi} = \arg\min_\pi \mathcal{L}_{match}$. Based on this assignment $\hat{\pi}$, our overall loss is defined as:

$$\mathcal{L} = \lambda_{loc}\mathcal{L}_{loc} + \lambda_{cls}\mathcal{L}_{cls} + \lambda_{saliency}\mathcal{L}_{saliency}, \tag{2}$$

where $\lambda_{loc}, \lambda_{cls}, \lambda_{saliency} \in \mathbb{R}$ are hyper-parameters balancing the three terms. $\mathcal{L}_{loc}$ combines an L1 loss and a generalized IoU loss (1D version as in (Rezatofighi et al., 2019)) between the prediction and ground-truth to measure the localization quality:

$$\mathcal{L}_{loc} = \lambda_{L_1}\mathcal{L}_1(\hat{\psi}^{\hat{\pi}(n)}, \psi^n) + \lambda_{gIoU}\mathcal{L}_{gIoU}(\hat{\psi}^{\hat{\pi}(n)}, \psi^n), \tag{3}$$

and $\mathcal{L}_{cls}$ is the cross entropy function to measure whether the action category is correctly classified or whether the prediction matches the query correctly. $\mathcal{L}_{saliency}$ directly borrows from Moment-DETR (Lei et al., 2021b) for query-dependent highlights detection. Hence, $\lambda_{saliency}$ is set to 0 for benchmarks without highlight detection.

## 4 EXPERIMENTS

This section presents our experimental setup, showcasing our pipeline's excellent performance. We first introduce datasets and metrics before reporting the comparison with the current state-of-the-art methods. We conclude the chapter with our thorough ablation study, which validates all our design choices, including noise level, decoder design, and query number.

### 4.1 DATASETS

Unless otherwise stated, we adopt the original datasets' data splits and benchmark on the test set.

**MAD (Soldan et al., 2022).** This recently released dataset comprises 384K natural language queries (train 280,183, validation 32,064, test 72,044) temporally grounded in 650 full-length movies for a total of over 1.2K hours of video, making it the largest dataset collected for the video language grounding task. Notably, it is the only dataset that allows the investigation of long-form grounding, thanks to the long videos it contains.

**QVHighlights (Lei et al., 2021a).** This is the only trimmed video dataset for the grounding task, constituted by 10,148 short videos with a duration of 150s. Notably, this dataset is characterized by multiple moments associated with each query yielding a total of 18,367 annotated moments and 10,310 queries (train 7,218, validation 1,550, test 1,542).

### 4.2 METRICS

**Recall.** Our metric of choice for the grounding task is Recall@$K$ for IoU=$\theta$. $K = 1$ if not specified.

**Mean Average Precision (mAP).** Following the literature, we compute the mAP metric with IoU thresholds $\{0.5, 0.7\}$. We also report the Average of mAP over multiple IoUs, denoted as AmAP.

### 4.3 IMPLEMENTATION DETAILS

Our training setting follows moment-DETR (Lei et al., 2021a). The input temporal spans get processed in the decoder blocks to predict their binary classes and the accurate boundaries. All the spans are used to train the binary classifier, but only the matched ones are applied to train the boundary predictor. Our algorithm is compiled and tested using Python 3.8, PyTorch 1.13, and CUDA 11.6. Notably, we have increased the number of encoder and decoder layers to 8, beyond which we observed saturation. We also use a fixed number of 30 queries (proposals) during both training and inference. To train our model, we use the AdamW (Loshchilov & Hutter, 2019) optimizer with a

**Table 1: Benchmarking of grounding methods on the MAD dataset.** We follow the methodology presented in (Barrios et al., 2023) and adopt a two-stage approach. In our work, we re-use the first stage implemented in (Barrios et al., 2023) and only work on the grounding (second stage) method. We report recall performance for baselines with (†) and without the first-stage model.

| Model | IoU=0.1 | | | | IoU=0.3 | | | | IoU=0.5 | | | |
|---|---|---|---|---|---|---|---|---|---|---|---|---|
| | R@1 | R@5 | R@10 | R@50 | R@1 | R@5 | R@10 | R@50 | R@1 | R@5 | R@10 | R@50 |
| Zero-shot CLIP (Soldan et al., 2022) | 6.57 | 15.05 | 20.26 | 37.92 | 3.13 | 9.85 | 14.13 | 28.71 | 1.39 | 5.44 | 8.38 | 18.80 |
| VLG-Net (Soldan et al., 2021) | 3.50 | 11.74 | 18.32 | 38.41 | 2.63 | 9.49 | 15.20 | 33.68 | 1.61 | 6.23 | 10.18 | 25.33 |
| Moment-DETR (Lei et al., 2021b) | 0.31 | 1.52 | 2.79 | 11.08 | 0.24 | 1.14 | 2.06 | 7.97 | 0.16 | 0.68 | 1.20 | 4.71 |
| DenoiseLoc (ours) | 1.06 | 4.07 | 6.75 | 20.07 | 0.86 | 3.34 | 5.44 | 15.67 | 0.57 | 2.17 | 3.50 | 9.73 |
| †Zero-shot CLIP (Soldan et al., 2022) | 9.30 | 18.96 | 24.30 | 39.79 | 4.65 | 13.06 | 17.73 | 32.23 | 2.16 | 7.40 | 11.09 | 23.21 |
| †VLG-Net (Soldan et al., 2021) | 5.60 | 16.07 | 23.64 | 45.35 | 4.28 | 13.14 | 19.86 | 39.77 | 2.48 | 8.78 | 13.72 | **30.22** |
| †Moment-DETR (Lei et al., 2021b) | 5.07 | 16.30 | 24.79 | 50.06 | 3.82 | 12.60 | 19.43 | 40.52 | 2.39 | 7.90 | 12.06 | 24.87 |
| CONE (Hou et al., 2022) | 8.90 | 20.51 | 27.20 | 43.36 | 6.87 | 16.11 | 21.53 | 34.73 | 4.10 | 9.59 | 12.82 | 20.56 |
| †DenoiseLoc (ours) | **11.59** | **30.35** | **41.44** | **66.07** | **9.08** | **23.33** | **31.57** | **49.90** | **5.63** | **14.03** | **18.69** | 29.12 |

learning rate of $1e-4$ and weight decay of $1e-4$. We train the model for 200 epochs and select the checkpoint with the best validation set performance for ablation. For evaluation, we compare our model with the state-of-the-art (SOTA) on the *test* split. We do not use large-scale pre-training as in (Liu et al., 2022), but we train our models from random initialization. For detailed experimental settings on each dataset, please refer to the *Appendix*. To extract video and text features, we follow the implementations described in the existing literature. Specifically, we use the MAD feature from the official release in (Soldan et al., 2022), the QVHighlights feature from (Lei et al., 2021a), and the language feature from GloVe (Pennington et al., 2014), as described in (Soldan et al., 2021).

## 4.4 COMPARISON WITH STATE-OF-THE-ART

Comparative studies are reported in this section. The best result is highlighted in bold in each table, while the runner-up is underlined.

**MAD.** Tab. 1 summarizes the performance of several baselines in the MAD dataset. For this evaluation, we follow the two-stage methodology introduced in (Barrios et al., 2023) and combine our DenoiseLoc with a Guidance Model. This model serves as a first-stage method that conditions the scores of the second-stage grounding methods to boost their performance. We report the performance of the baselines with and without the Guidance Model, highlighting the former with the † symbol. Note that CONE (Hou et al., 2022) does not use the same first stage, yet it implements the same concept with different details.

First, let's analyze the second stage's performance (rows 1-4). We can observe that proposal-based methods such as Zero-Shot CLIP and VLG-Net (Soldan et al., 2021) offer much stronger performance with respect to the proposal-free approaches Moment-DETR (Lei et al., 2021b) and DenoiseLoc. This observation is congruent with many other findings in the literature, even beyond the localization tasks video. In this case, the strongest challenge is constituted by the long-form nature of MAD, which naturally yields many false positive predictions. As it turns out, proposal-free methods are particularly susceptible to this phenomenon. Nonetheless, the combination of guidance and grounding models is able to significantly boost these methods' recall by removing false-positive predictions. In fact, our DenoiseLoc obtains the highest metric for most configurations, with large margins (between 9.6% and 34.4% relative improvements) against the runner-up methods.

**QVHighlights.** We find that our method's good behavior translates to other datasets with very different characteristics, as in QVHighlights (Lei et al., 2021a). Recall and mAP performance are presented in Tab. 2, where DenoiseLoc obtains the best performance for all metrics with relative improvements ranging from 3.4% and 14.0%. Note, that we strive for fair comparisons; therefore, we do not report Moment-DETR and UMT performance when pretraining is used. Also, our method surpasses the concurrent work DiffusionVMR (Zhao et al., 2023), and we only run one denoising step. Additional details can be found in (Liu et al., 2022).

## 4.5 ABLATIONS AND ANALYSIS

We design several ablations and analysis studies to probe our method and devise meaningful takeaways that can help in identifying fruitful future directions.

**Table 2: Benchmarking of grounding methods on the *test* split of QVHighlights dataset.** Note that our method surpasses the concurrent work DiffusionVMR (Zhao et al., 2023), and we do only one denoising step.

| Model | R@1 | | mAP | | |
|---|---|---|---|---|---|
| | @0.5 | @0.7 | @0.5 | @0.75 | Avg. |
| MCN (Hendricks et al., 2017) | 11.41 | 2.72 | 24.94 | 8.22 | 10.67 |
| CAL (Escorcia et al., 2019) | 25.49 | 11.54 | 23.40 | 7.65 | 9.89 |
| XML (Lei et al., 2020) | 41.83 | 30.35 | 44.63 | 31.73 | 32.14 |
| XML+ (Lei et al., 2021a) | 46.69 | 33.46 | 47.89 | 34.67 | 34.90 |
| Moment-DETR (Lei et al., 2021b) | 52.89 | 33.02 | 54.82 | 29.40 | 30.73 |
| UMT (Liu et al., 2022) | 56.23 | 41.18 | 53.83 | 37.01 | 36.12 |
| DiffusionVMR (Zhao et al., 2023) | 61.61 | 44.49 | 61.55 | 40.17 | 39.92 |
| QD-DETR (Moon et al., 2023) | **62.40** | 44.98 | **62.52** | 39.88 | 39.86 |
| DenoiseLoc (ours) | 59.27 | **45.07** | 61.30 | **43.07** | **42.96** |

**Table 3: Ablation study on the number of queries.** When we have fewer queries, adding noise helps the model quickly converge to a stable state, resulting in improved performance. However, increasing the query number will slow down the convergence rate because of the increased negatives.

| Query Number | Denoise Training | Epoch Number | | | | |
|---|---|---|---|---|---|---|
| | | 5 | 10 | 20 | 40 | 80 |
| 8 | ✗ | 12.60 | 26.30 | 36.44 | 39.65 | 41.46 |
| | ✓ | 11.84 | 30.62 | 39.73 | 41.59 | 42.18 |
| 16 | ✗ | 4.92 | 26.80 | 38.35 | 41.82 | 42.22 |
| | ✓ | 12.26 | 31.45 | 39.34 | 42.52 | 43.78 |
| 32 | ✗ | 7.91 | 25.37 | 39.07 | 41.38 | 42.27 |
| | ✓ | 4.16 | 22.36 | 38.42 | 43.98 | 44.11 |

**Table 4: Ablation study on noise ratio.** A small noise leads the model to underfit, while a large noise reduces the prediction variance and performance. Note that dB (decibel) is expressed in logaritmic scale. $\text{NoiseRatio}_{dB} = 10 \log_{10} \frac{P_{\text{noise}}}{P_{\text{signal}}}$, $P$ is the average of the squared values.

| Noise Ratio (dB) | (-20,-10] | (-10,0] | (0,10] | (10,20] |
|---|---|---|---|---|
| Avg. mAP (avg) | 43.71 | 44.66 | 44.22 | 42.89 |
| Avg. mAP (std) | 1.64 | 0.86 | 0.56 | 0.40 |

**Noise level.** Tab. 4 investigates the impact of varying noise levels (measured by noise-to-signal ratio). Specifically, we gradually increased the noise level from -20dB to 20dB to assess the model's performance. Note, dB refers to decibel. Our findings suggest that the model's performance follows a parabolic pattern as the noise level increases, with the best results achieved 44.66% mAP in the (-10, 0] range. Additionally, it also implies that a noise level that is too small will risk the model to overfit, resulting in unstable performance and higher prediction variance. Conversely, a noise level that is too large will lead to underfitting, reducing prediction variance but causing a corresponding decrease in model accuracy.

**Decoder design.** Tab. 5 illustrates the effect of various decoder designs with and without denoising training. Several observations are concluded: **1.** Our model, built upon moment-DETR, assesses different ways to combine proposal and query features in Rows 1 and 2. The first row shows that using the average of the feature to represent the action instance instead of Dynamic Conv. also improves performance. **2.** Row 3 and Row 4 show when self-attention is applied, significant improvements are observed, which suggests the proposal-proposal interaction plays an important role in global message passing. **3.** Compared with the performances without denoising training, nearly all the metrics improved once the noisy temporal spans participated in model training. The improvements over non-self-attention models (Row 1, 2) are more significant, although we don't introduce any extra computation in the decoder. We also find the best model performance is achieved by enabling all the modules, especially Dynamic Conv. This is because the Dynamic Conv. operation is more sensitive to the temporal feature of the proposals, and our denoising training, by jittering the proposal boundaries, can bring more augmentations in the target data domain.

**Table 5: Ablation study on our model architecture on the *val* split of the VQ-highlight dataset.** When self-attention is applied, we observe significant improvements, which suggests the proposal-proposal interaction plays an important role in global message passing. Also, nearly all the metrics improved once the noisy temporal spans participated in model training.

| Dynamic Conv. | Self Att. | *w.o.* Denoise Training | | | *w.* Denoise Training | | |
|:---:|:---:|:---:|:---:|:---:|:---:|:---:|:---:|
| | | IoU@0.5 | IoU@0.7 | AmAP | IoU@0.5 | IoU@0.7 | AmAP |
| ✗ | ✗ | 54.1 | 39.0 | 32.3 | 57.2 | 44.1 | 38.5 |
| ✓ | ✗ | 46.1 | 34.2 | 28.6 | 55.9 | 43.4 | 41.6 |
| ✗ | ✓ | 57.2 | **45.0** | 41.1 | 57.8 | 45.3 | 40.8 |
| ✓ | ✓ | **58.1** | 44.7 | **43.0** | **59.9** | **45.9** | **44.6** |

**Query number.** Tab. 3 presents the performance comparison of models trained on different epochs with varying numbers of queries. The first two blocks suggest that after adding noise, the model can quickly converge to a stable state, resulting in improved performance. This observation is consistent with the results shown in Fig. 4. Increasing the number of queries from 8 to 16 and then to 32, adding noise at the beginning helps the model to converge faster and perform better. However, as the number of queries exceeds a certain threshold, the convergence speed slows down, and performance deteriorates. This may be due to the increasing number of negative samples resulting from one-to-one assignments performed by the Hungarian matcher, which makes it harder for the model to learn how to denoise and causes decreased performance. This finding also highlights that our model can achieve excellent performance with a small number of predictions, rendering NMS unnecessary.

## 4.6 FURTHER DISCUSSION

**Table 6: Adapting our denoising model to a diffusion framework.** No benefit is found when transforming our denoising approach to a diffusion one.

| Steps | 1 | 2 | 4 | 8 | 64 | 512 |
|:---:|:---:|:---:|:---:|:---:|:---:|:---:|
| Avg. mAP | **44.26** | 43.85 | 43.04 | 43.61 | 43.57 | 42.99 |
| mAP@0.5 | **62.08** | 61.08 | 60.21 | 60.89 | 60.66 | 59.71 |
| mAP@0.7 | **44.96** | 43.65 | 43.04 | 44.74 | 44.35 | 43.20 |

**DenoiseLoc *v.s.* DiffusionLoc.** Our model can be viewed as a conditional diffusion model when time embedding (parameter governing the diffusion process) is also included in the dynamic convolutional layer. In this case, the input of our model is a temporal span of a certain level of noise, and the output is the offset of the proposal to the target activity localization. As an extension of Tab. 4, we design a DDPM (Ho et al., 2020)-like training protocol (Chen et al., 2022) to optimize our model, dubbed as **DiffusionLoc**. Specifically, in each training iteration, we random sample an integer $t \in \{1, \cdots, T\}$, and decide the noise level $\sigma$ based on it. Then the following process is the same as **DenoiseLoc**, we denoise the ground truth action location to train the network and prediction the denoised proposal in model inference. We investigate $T \in \{1, 2, 4, 8, 64, 512\}$ and report the model performance in Tab. 6. More discussion can be found in Appendix E.

Surprisingly, we don't observe any evident performance gain of DiffusionLoc from DenoiseLoc, which achieves $44.66 \pm 0.86$ on the same validation split. One potential reason is that knowing the noise level does not help the model to localize activities more precisely, and denoising training is already an effective method in temporal activity localization. This motivates why our method surpasses the concurrent work DiffusionVMR (Zhao et al., 2023) in one single denoising step.

## 5 CONCLUSION

We propose DenoiseLoc, an encoder-decoder model, which introduces a novel boundary-denoising training paradigm to address the challenge of uncertain action boundaries in video activity localization. DenoiseLoc captures the relations within and across modalities in the encoder and progressively refines learnable proposals and noisy ground truth spans in multiple decoder layers. Our boundary-denoising training jitters action proposals and serves as an augmentation to guide the model on predicting meaningful boundaries under the uncertainty of initial noisy spans. Extensive experiments to demonstrate the effectiveness of DenoiseLoc, achieving state-of-the-art performance on several datasets, including the large-scale MAD dataset.

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

# A   Appendix

## A.1   Limitations and Ethics Concerns

**Limitations:** A common limitation of deep learning-based computer vision solutions for video understanding is the potential for overfitting. This is driven by the adoption of very large models and sometimes small datasets. In this work, we address this concern by evaluating the benefit of our approach across multiple datasets, some of which are of massive scales, like MAD. Yet, our work presents itself with other limitations. In fact, throughout the experimental section, we mentioned how our performance for high recall (i.e., R@5) might not achieve state-of-the-art results. We hypothesized that this phenomenon is due to the Hungarian matcher that is adopted in this work and several others (Tan et al., 2021). Such an algorithm assigns the ground truth to the best prediction, allowing us to compute the loss only for R@1 and not optimize for multiple predictions. We believe researchers should focus on improving this aspect of the current set-prediction approaches. We leave the solution to this limitation to future works.

**Ethics Concerns:** As the performance of video understanding algorithms gradually improves, such techniques, publicly available for research purposes, might be adopted for illicit surveillance purposes, raising concerns about privacy and potential misuse. Our work is released to the public in accordance with the limitation of the MIT license. Any misuse of the information provided in this document is to be condemned and apprehended.

## A.2   Experiment settings

This section presents the detailed experimental setup for each dataset evaluated in the main paper.

**MAD (Soldan et al., 2022).** We used this dataset to evaluate our method under the long-form video setting. To have a fair comparison with the literature, we follow a coarse-to-fine two-stage method. We only use the DenoiseLoc prediction as a finer prediction in a short window that is predicted by (Barrios et al., 2023). Also, we reduce the number of queries from 30 to 14 and do not find any performance drop. In addition, since MAD does not have snippet-wise annotation that could be used as saliency loss in Eq. (2), we randomly sample 3 snippets inside the temporal span annotation as positives to help the encoder's convergence.

**QVHighlights (Lei et al., 2021a).** Our training process follows precisely the description of our Sec. 4.3. To receive the test set performance, we submit our prediction to the official server hosted on CodaLab.

## A.3   Training Efficiency

DenoiseLoc is an encoder-decoder model that incorporates a temporal span denoising task during training. In Fig. 4, we demonstrate that our denoising training yields a fast converging model.

## A.4   Visualizations

We visualize the predictions from decoder layers 1, 2, 4, and 8 of a video example in Fig. 5. We use colored arrows to denote predicted temporal spans (the darker the color, the higher the model confidence). We also filtered the predictions if the confidence is less than $0.2$. As the decoder goes deeper (bottom to up), the proposal boundaries become more precise, and the ranking confidence scores become more accurate. In the meantime, the redundant proposals are also suppressed.

## A.5   Denoise training on decoder (denoiser)

**Training as a denoiser**. The proposed denoising training can be viewed in two ways. (1) *Augmented training set*. Besides the learned proposal span, we generate random spans around the ground-truth location. Those new spans can accelerate the decoder's convergence and make our training more robust. (2) *Individual module*. Since the generated spans and learned spans do not interact in the transformer decoder layers, they can be studied as in two different modules independently, while the two modules share the same model architecture and weights. Therefore, the individual module, *e.g.* denoiser, works similarly to a regularizer.

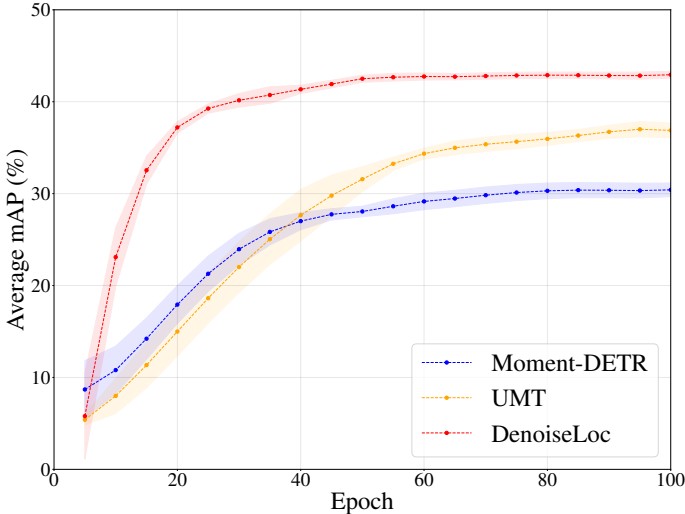

**Figure 4:** Our method has faster convergence and better performance compared to other state-of-the-art methods on the QV-Highlights dataset.

**Query:** *Man wears a bandana around his face while out and about.*

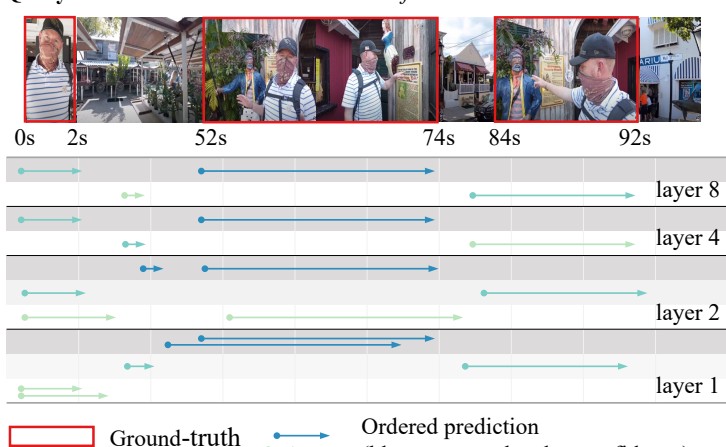

**Figure 5: Qualitative Results on QV-Highlights.** A deeper color means the proposal's confidence score is higher. Our method progressively denoises the action boundaries and removes the redundancies. See Sec. A.4 for more details. Best viewed in color.

**Spans in inference**. Although the denoiser module is disabled during inference as the ground-truth information is not available anymore, we discover that replacing the learnable spans with random spans also gives a competitive performance, see Tab. 7. The average performance (*e.g.* AmAP in *all* setting) stays almost the same, but using random spans gives more advantages under a more strict metric IoU=0.7 and is also good at middle/short activities, shown in the last two columns. Therefore, we can deduce the learned temporal spans are good at long activities over lenient constraints, while random spans have a higher chance of predicting precise action boundaries over short activities.

**Uncertainty in inference**. Following the random span setting in inference, another good property of DenoiseLoc is that the uncertainty of model predictions can be measured by generating random spans with different random seeds. To explore this, we change the seed from 0 to 9, run inference, and visualize the predictions on the same video from all runs in Fig. 6 and Fig. 7. The ground-truth spans in red are in the top subplot, and the predictions from ten different seeds are in the bottom. In each subplot of the predictions, the one with higher confidence is higher and of darker color. Predictions with confidence scores less than 0.2 are removed. The unit on the x-axis is per second.

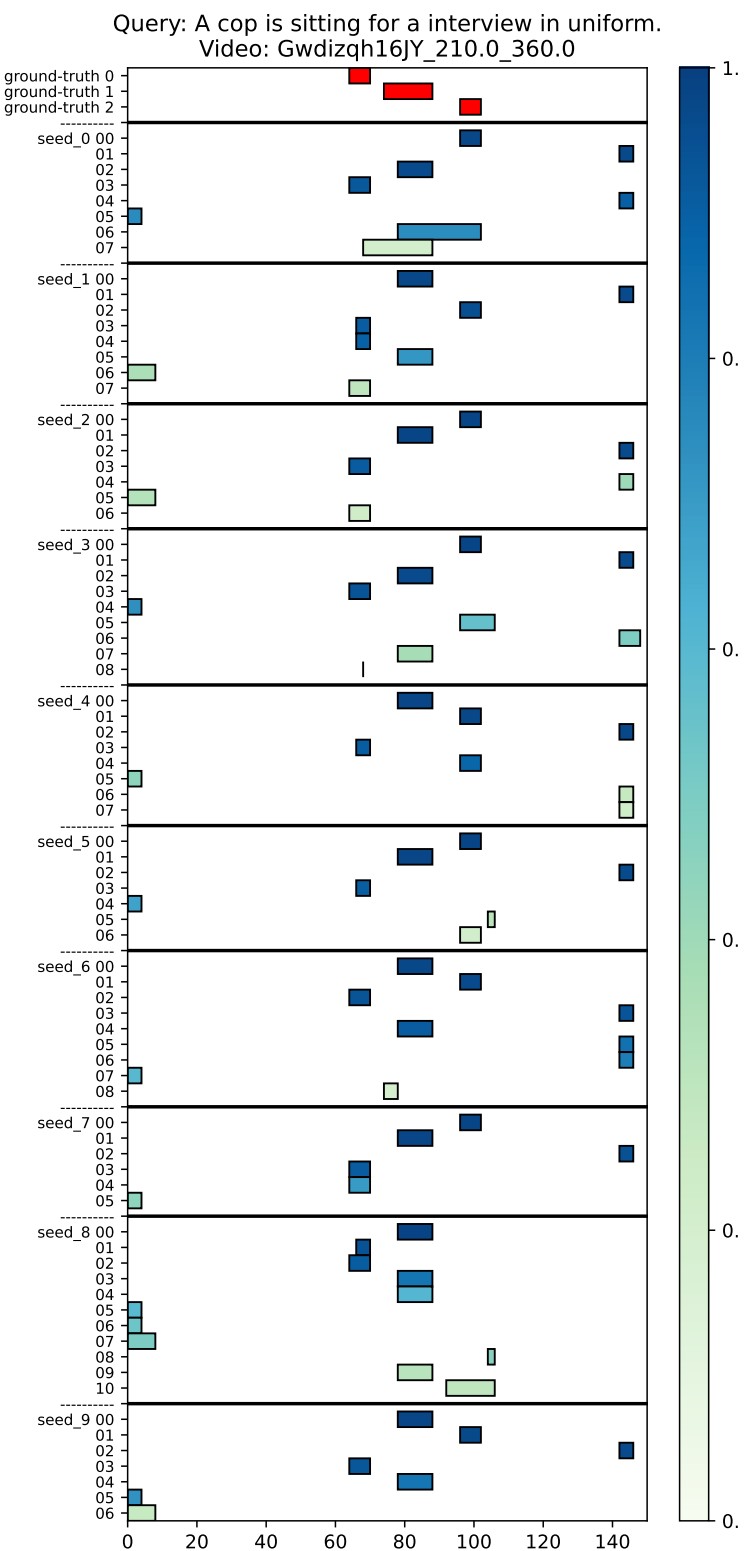

**Figure 6: Visualization with ten random seeds.** Almost all subplots have correct predictions, although orders are different. Link.

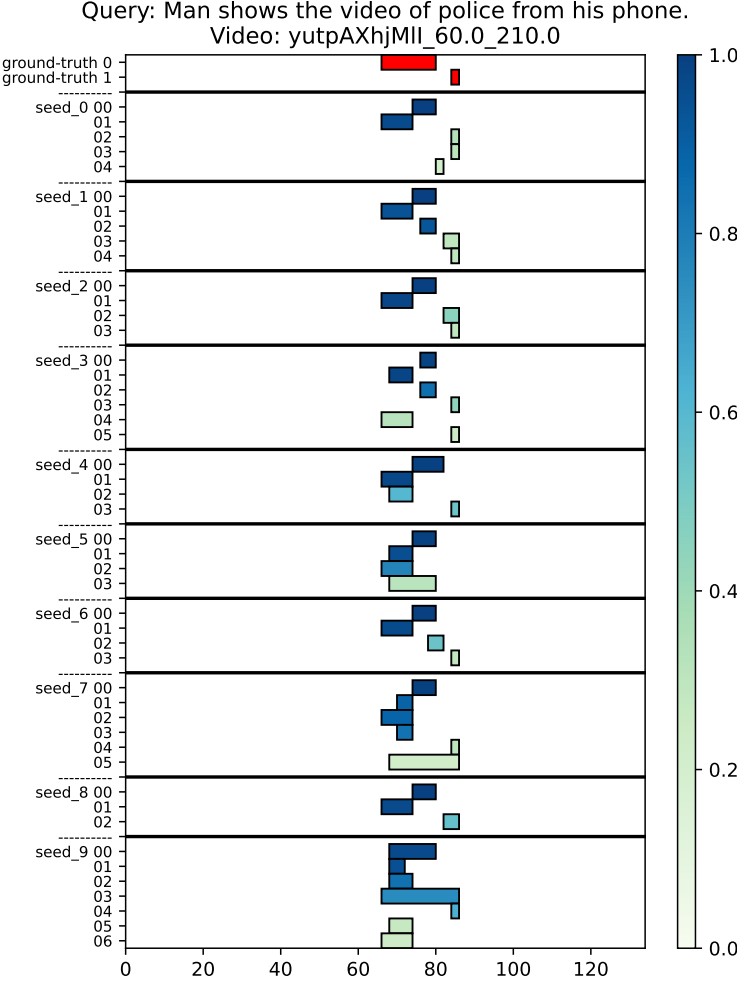

**Figure 7: Visualization with ten difference random seeds.** The model is more sensitive to the random seed in this case. Link.

**Table 7: Comparison of different spans during inference.** Replacing the learnable spans with random ones also give competitive performance.

| span | Top 1 Recall | | Average mAP | | | |
| | @0.5 | @0.7 | all | long | middle | short |
| --- | --- | --- | --- | --- | --- | --- |
| learned | **60.52** | 45.23 | 44.41 | **52.67** | 45.10 | 11.68 |
| random | 59.29 | **46.84** | **44.51** | 50.82 | **45.75** | **12.10** |

Fig. 6 visualizes our results to the query "*A cop is sitting for an interview in uniform.*" We observe that almost all subplots have correct top-3 predictions, although their order could differ. Experiment with seed 8 misses a target probability because all the random spans are far away from this location. Thus we don't find any more predictions after timestamp 110 (x-axis) for seed 8.

Fig. 7 is more challenging according to our predictions because the model is more sensitive to the random seed. We observe both the central location and the duration of the top-1 prediction vary a lot over the 10 runs, while it seems that only the seed-9 experiment gives a correct answer. However, if we take a closer look at the video, there is a *camera-change* from 2:13-2:15 (equivalent to 73-75 in our x-axis). Therefore, our model predicts it as two separate activities, but both are relevant to the query, "*Man shows the video of police from his phone.*"

## A.6 SUPPLEMENTARY TO OUR DIFFUSION VARIANT

**DiffusionLoc** A time embedding can be incorporated into the dynamic convolutional layer in the decoder. We name this variant of our model as DiffusionLoc. To be more concrete, given an integer $t$, we project its sinusoidal position embeddings by multi-layer perception to produce scale and shift vectors, denoted as $s_1$ and $s_2$. Given the original output $X$ after the FC layer, our time-conditioned output is $(s_1 + 1)X + s_2$. Note that we have used $\psi$ as spans, so the decoder can be formulated as

$$\psi_{pred} = Dec(\psi_{noise}^{(t)}, t, Memory, \Theta)$$

, where $Memory$ is the output from the encoder and $\Theta$ is the learnable parameters of the model.

The training phase of DiffusionLoc is pretty much the same as DenoiseLoc, while the difference is that the noise level is controlled by $t$, randomly sampled in $\{1, 2, \cdots, T\}$ in each iteration and each video, *i.e.*,

$$\psi_{noise}^{(t)} = \sqrt{\bar{\alpha}_t}\psi_{gt} + \sqrt{1 - \bar{\alpha}_t}\vec{\epsilon},$$

where $T = 1000$, the choice of $\bar{\alpha}_t$ follows DDPM (Ho et al., 2020), and $\vec{\epsilon}$ is the noise term as defined in our DenoiseLoc.

The ideal inference of DiffusionLoc is to take from $T, T-1, \cdots$ until 1. In each step, we estimate the noisy spans at the noise level at $t$, and the decoder will predict the activity intervals from the estimated spans. Mathematically, each step can be shown as,

$$\psi_{pred}^{(t)} = Dec(\tilde{\psi}_{noise}^{(t)}, Memory, t)$$
$$\tilde{\psi}_{noise}^{(t)} = \sqrt{\bar{\alpha}_t}\psi_{pred}^{(t+1)} + \sqrt{1 - \bar{\alpha}_t}\vec{\epsilon}.$$

In practice, we observe the predicted negative spans degenerate quickly and cannot be recovered to a proper random distribution, while the predicted positive spans also have the risk of being negative after the *diffusion* process. Therefore, it is best for us to have only one step in inference. However, given a converged DenosieLoc model, we can always progressively boost the model performance by using different sets of initial spans and ensemble the predictions from each run. We would like to gain attention from the community and leave this study and follow-up works from DenoiseLoc.

## A.7 COMPUTATIONAL BLOCKS

In this section, we provide additional information and visualizations regarding the RoI alignment (Align1D) (Xu et al., 2020) and DynamicConv (Sun et al., 2021) blocks.

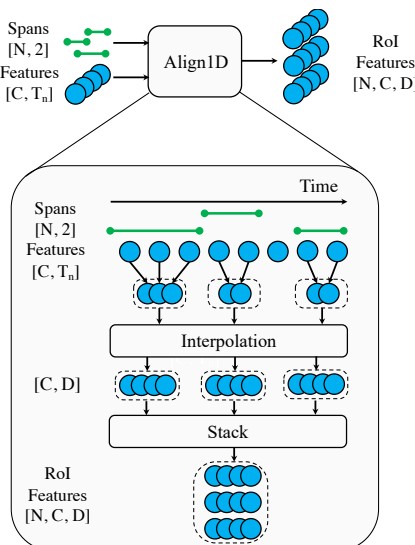

**Figure 8: RoI Alignment (1D).** See text for details.

**RoI Align** Figure 8 illustrates the workflow of the RoI Align block in one dimension case. This block, termed Align1D or RoI Align 1D is borrowed directly from (Xu et al., 2020) and builds on the original RoI Align 2D introduced in (He et al., 2017).

This block accepts in input a set of visual features (shape $[C, T_n]$), where $C$ is the feature channel and $T_n$ is the number of features with $n \in \{1, \ldots, N\}$, and a list of temporal spans (shape $[N, 2]$). In our implementation, the features are the memory from the transformer encoder (see Section 3 and Fig. 3). For the $i$-th temporal span, we first identify those elements of the features that fall within the span's temporal endpoints and group them together. Each group undergoes an interpolation operation to produce a fixed number ($D$) of features regardless of the input feature number, resulting in a new tensor with shape $[C, D]$, as shown in the figure. The outputs from all groups are finally stacked together to produce the output feature of dimension $[N, C, D]$. The use of this block allows us to obtain a fixed dimensional representation feature for each temporal span.

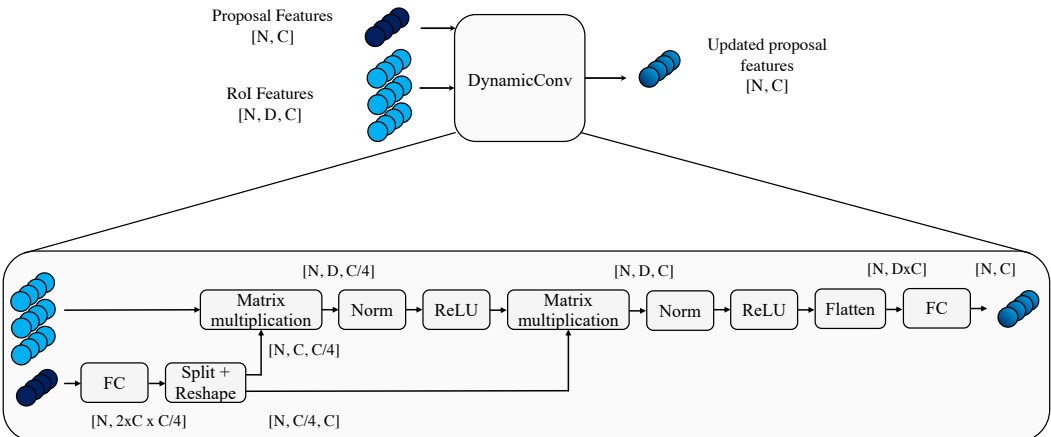

**Figure 9: DynamicConv.** See text for details.

**DynamicConv** Figure 9 illustrates the workflow of the DynamiConv block. This block is borrowed directly from (Sun et al., 2021), which was used in the object instance segmentation task.

DynamicConv accepts in input two sets of features. In our implementation, we feed the RoI features, the output of the RoI Align block, and the proposal features.

DynamicConv is a special type of Dynamic Filter (Jia et al., 2016) and Conditional Operation (Tian et al., 2020) proposed for the detection problem (Sun et al., 2021). It improves RoI features from a given proposal features and outputs an enhanced feature of all the temporal spans. In our implementation, the RoI features are extracted from the temporal spans and *memory* by Align1D. Besides, the proposal features are originally from learnable embeddings in the first decoder block and from the output of previous DynamicConv layers in the following decoder blocks.

To be more specific, DynamicConv projects the proposal feature ($[N, C]$) into two parameter groups with increased dimensionality, $[N, C, C//4]$, $[N, C//4, C]$. Then, the RoI feature shaped in $[N, D, C]$ is multiplied by them sequentially with non-linear functions to gain instance-level information. The result shaped in $[N, D, C]$ is then flattened and projected as a new $[N, C]$ feature. This feature is used for boundary denoising and improving the RoI feature of the next decoder block.

