# OpenReview forum: "Boundary Denoising for Video Activity Localization"
_ICLR.cc/2024/Conference — ICLR 2024 poster_

### Official Review · Reviewer_61UD · 2023-10-25

**Soundness:** 1 poor
**Presentation:** 2 fair
**Contribution:** 2 fair
**Rating:** 3
**Confidence:** 3

**Summary:**

The authors propose a method for boundary denoising to tackle video activity localization. A model architecture DenoiseLoc is proposed, together with a boundary denoising training method. The authors argue that a single step denoising is better than the diffusion process with multiple steps. Experiments show some improvement over the previous state-of-the-art.

**Strengths:**

The experimental results show some improvements over the previous state-of-the-art.

**Weaknesses:**

Most importantly, the method part is not clear. It is rather hard to follow through most of the written parts. Figure 2’s caption also does not provide a clear overview of the proposed method and the novel aspects.


In Figure 2, it is rather confusing what the pipeline actually is. For instance, the ground truth span/noise injection part should definitely not be part of the inference pipeline. So, it is not clear what is done during the inference process.


Suddenly, in 3.2.2, the dynamic convolution is used without much explanation. Why is it important to the proposed design? What is the dynamic convolution exactly doing, and why no other design can be used? It is not well motivated.


The boundary denoising training part in 3.2.3 is not clear at all. How the method works, what loss is used, where the loss is applied, and why it is designed this way is not clear. Why do we need to divide into two different groups? How does the model use both of them during training?

Importantly, since boundary denoising has been widely explored, what are the further insights that make the proposed method more effective than previous works? This has not been clearly expressed.




Experimentally, there are also some parts not well established.

Most importantly, it is very strange to me why adding diffusion will lead to performance drops. Furthermore, the more steps used, the worse the performance seems to get. This is totally different from what is usually observed in many diffusion-based works (for generation and for prediction tasks). Usually, the benefit of using a single step is only for efficiency purposes. Furthermore, the given reason is also not convincing. It would be good if the authors provide a lot more details about how diffusion is used, and more qualitative/quantitative evidence to substantiate this claim, since it is quite a strong and counterintuitive claim.


It seems that more ablations are required for various designs, for example the various designs in denoising training. But, currently the method is too unclear for me to suggest concrete settings.







Note that there are some mistakes with the spelling/formatting. This does not affect the score. Some examples are:

Pg 2 bottom: citation formats are mixed up, and all the authors of the papers are listed in the citation.
Pg 1 and Pg 2: “QV-Hightlights”
Pg 9: “prediction the denoised proposal”, “more discussion”

Throughout, please standardize as DenoiseLoc (instead of denoiseloc at several parts).

**Questions:**

Apart from the above concerns, some other specific questions are below.


1)	Could the authors provide the time gains from using a single denoising step against multiple?
2)	Could the authors provide the model size and time gains as compared to previous state-of-the-art methods?
3)	In table 4, when the dB measure for noise is used, what exactly does it mean in this context?

---

> ### Author Response · Authors · 2023-11-21
>
> **General Answer**
>
> We sincerely thank the reviewer for his time dedicated to assessing our work. While we understand that our manuscript may have been challenging to comprehend, we appreciate the reviewer's feedback and have attempted to address the confusion in the revision. We would like to point out that other reviewers have all expressed positive feedback regarding the presentation of the idea. Nonetheless, we are very receptive to constructive and concrete suggestions that can improve clarity and readability and improve our submission.
> Follow a general answer tackling the presented weaknesses. In the next comment box, we have a more discussion on Denoising vs Diffusion, and also answer the specific questions asked by the reviewer. Note that all changes to the manuscript are highlighted in blue in the revised version for the convenience of the reviewers.
>
> **1- Figure 2.** We revised the caption of Figure 2. The reviewer can verify the change in the updated manuscript. We report here the updated text for ease of reference.
>
> “Our proposed model consists of three main components: Encoder, Decoder, and Denoising Training. The input video and text are transformed into video and text embeddings by means of pretrained networks, then concatenated and fed to the Encoder module (left). The output features from the Encoder, termed as memory, are subsequently routed to the Decoder. During training, the Decoder receives in input a set of learnable embeddings, ground truth spans with additional artificial noise, and a set of learnable spans. The objective of the decoder is to enrich instance-level features computed based on the two sets of spans and refine (via denoising) the provided ground truth spans. Note that the GT spans are not used during inference. ”
>
> We would like to clarify that the pipeline refers only to the training phase. The behavior of the model at inference time is presented in the manuscript in Section 3.2, where we explain that the Denoise is disabled during inference. The decoder is the module that refines the learnable temporal spans used as predictions. Further details are present in Section 3.2.3, where we detail the boundary denoising training.
>
>
> **2- DynamicConv.** We would like to clarify that DinamicConv is not a contribution of our work. It was in fact, introduced in Sparse-R-CNN [1] and was borrowed as is. We adopt this computation block in pursuit of a good Decoder design and ablate several configurations in Section 4.5. Therefore, DynamicConv is not a necessary block for the correct functioning of our method, yet it is clearly shown to help achieve the best performance in the reported ablation in Table 5, as the module operation is more sensitive to the temporal feature extracted from the temporal span. Nonetheless, we would like to reinforce that our proposed method brings improvements even when such a block is not utilized. See Tab 5, row 1.
>
> We are happy to provide further clarification if the reviewer deems it necessary.
>
> [1] Sun, Peize, et al. "Sparse r-cnn: End-to-end object detection with learnable proposals." Proceedings of the IEEE/CVF conference on computer vision and pattern recognition. 2021.
>
>
> **3- Section 3.2.3.** We reorganized this section based on the reviewer's feedback. Please directly check it from our updated version.
>
>
> **4- Missing literature.** We would like to kindly invite the reviewer to provide us with a list of relevant literature papers exploring the temporal boundary denoising problem in video localization tasks that were available before the submission of this manuscript. We believe he might find this literature lacking, reason why we decided to focus our effort in this direction. To the best of our knowledge, the relevant literature is included, however, we acknowledge that one might always miss some recent and promising works. We are happy to include missing literature in our related work and compare performance where possible.
>
> **5- Evaluation protocols.** We would like the reviewer to point out the non-standard evaluation protocols involved in this manuscript. We base our work and evaluation code on the Moment-DETR repository. Please also note that the test-split annotation is not open to the public, so we submit our predictions to their server for a fair comparison. We are, however, open to further entertaining a discussion regarding this topic with the reviewer to clarify any doubts.
>
>
> **6- Additional ablations.** We kindly ask the reviewer to provide actionable suggestions on the ablation designs. This would greatly help us in narrowing down the relevant experiments, helping us in designing and analyzing the results against potential expectations the reviewer might have.

---

> > ### Author Response · Authors · 2023-11-21
> > **Denoising vs Diffusion, and Answer to Questions**
> >
> > **Denoising vs Diffusion.**
> >
> > We agree with the reviewer that our discovery is not in line with the current trends in diffusion-based work. However, the reviewer might agree with us that our solution space (temporal endpoints) has a much smaller support set (i.e., dim=2) with respect to the solution space of image generation works (image space output). (e.g. dim=512x512, or even 1024x1024)
> > our system is tasked with predicting only two real values, while most applications of diffusion deal with much more complex outputs. Therefore, it is plausible, and verified by our work, that the adoption of denoising vs diffusion can provide favorable results. Additionally, the increased inference efficiency is a bonus one should be aware of.
> >
> > Our claim and contribution to the community is to showcase that a single denoising step is sufficient for achieving high performance in video localization, suggesting that denoising training is already an efficient and effective method for this task.
> >
> > However, we do not deny the possibility that diffusion may be appropriate for detection tasks. We hope that our work will spark further discussion and research in the field of diffusion-based models for recognition-related tasks such as object detection and video localization.
> >
> > **Q1- Time cost of multiple denoising steps.** We compare the running time on different denoising steps in ms in the table below. This localization running time is measured only from the video feature being fed into the localization model until the model gives the output, averaging over 1000 runs. It doesn't include video encoding processes such as video frame extraction and video feature extraction.
> >
> > | Denoising Steps      | 1     | 2 | 4 | 8 | 64 | 512 |
> > |----------------------|-------|---|---|---|----|-----|
> > | Localization Runtime | 19.47 | 34.02  |  43.55 |  89.92 |  718.75  |  5.55k   |
> >
> > **Q2- Model size and time cost with respect to art.** We compare the running time and peak GPU memory of our method and Moment-DETR in the table below. The peak memory is the max value reported from the NVIDIA system management interface (i.e., `nvidia-smi`). The pipeline running time is from the video beginning to load to the localization being predicted, averaging over 100 runs. The localization running time is as defined in Q1. In the last column, we also report the model performance, average mAP, on the val-split of QV-Highlights.
> >
> > | Method                      | Peak GPU memory (MB) | pipeline runtime (ms) | localization runtime (ms) | Performance (AmAP) |
> > |-----------------------------|----------------------|-----------------------|---------------------------|--------------------|
> > | Moment-DETR      | 1843                 | 984.84                | 5.22                      | 32.20              |
> > | Ours (w. 8 decoder layers) | 2145                 | 1057.96               | 19.47                     | 45.29              |
> > | Ours (w. 1 decoder layers) | 1851                 | 936.72                | 5.24                      | 38.48              |
> >
> > The peak GPU memory usage of our model (8 layers) increased from 1843 MB to 2145 MB. The run time of our localization model (8 layers) runs a bit slower than the baseline (19.47 ms vs 5.22 ms)
> >
> > Note:
> > (1) Our model in 1 decoder layer setting takes similar running time and GPU memory to the baseline but reaches higher performance
> > (2) Compared with the entire video localization pipeline, the major latency is on video frame extraction and frame/language feature extraction, while the localization process takes less than 1% of the whole latency.
> >
> > The values of model size and time cost of other start-of-the-art methods may vary from Moment-DETR. However, since the main computation bottleneck in video localization is to compute the video/text feature before running the localization model, we consider this extra computation overhead to be negligible.
> >
> >
> > **Q3- Table 4 notation clarification (dB).** The notation dB refers to decibel, a measure of signal-to-noise ratio commonly used in engineering fields, especially those dealing with signal processing. Note that decibel is often reported in logarithmic scale, as is the case in Table 4. We have updated the table caption and the corresponding section for improved clarity.

---

### Official Review · Reviewer_6dfr · 2023-10-27

**Soundness:** 4 excellent
**Presentation:** 3 good
**Contribution:** 3 good
**Rating:** 8
**Confidence:** 4

**Summary:**

This paper proposed an encoder-decoder model, namely DenoiseLoc, for video activity localization. DenoiseLoc introduces a boundary-denoising paradigm to address the challenge of uncertain action boundaries. DenoiseLoc leverages across modalities in the encoder and progressively refines learnable proposals and noisy ground truth spans in decoder layers. Extensive experiments on standard benchmarks demonstrate the effectiveness of the proposed DenoiseLoc.

**Strengths:**

- A novel boundary-denoising paradigm is proposed to address the challenge of uncertain action boundaries in video activity localization task.
- Extensive experiments on standard benchmarks demonstrate the effectiveness of the proposed DenoiseLoc.
- It is interesting to find that satisfactory performance can be achieved with very few proposals and very few denoising steps.

**Weaknesses:**

-  Lack of visual analysis. It would be helpful to understand the properties of the proposed method if some cases can be visually analyzed.
- "DenoiseLoc" and "denoiseloc" are used interchangeably, which confuses readers. It is recommended that all be changed to "DenoiseLoc".

**Questions:**

Please refer to Weaknesses for more details.

---

> ### Author Response · Authors · 2023-11-21
>
> **General Answer**
>
> We sincerely thank the reviewer for his time dedicated to assessing our work, acknowledging its relevance, and providing suggestions on how to strengthen it. We are excited the reviewer has found our work technically robust and well-presented. Moreover, the reviewer's belief that our work makes a valuable contribution to the community is especially gratifying.
>
> In response to the feedback, we have revised our manuscript to ensure consistency in the naming of our model throughout the document. Note that all our changes are highlighted in blue in the revised manuscript for the convenience of the reviewers.
>
> Furthermore, we are very receptive to the idea of including additional visualizations and would welcome any specific suggestions the reviewer may have in this regard. Currently, we provide an Illustration of the boundary denoising process in Figure 1, where we showcase that we can effectively achieve more accurate temporal boundaries through the cascade of denoiser blocks. This figure showcases the efficacy and cascade effect of the denoising tower. Additionally, in the Appendix (Figure 5), we offer visual representations of the solution process for a specific video-query pair, visually illustrating the task.
>
> We are happy to provide additional visualizations following the reviewer's suggestions.

---

### Official Review · Reviewer_uvci · 2023-10-28

**Soundness:** 3 good
**Presentation:** 2 fair
**Contribution:** 3 good
**Rating:** 6
**Confidence:** 5

**Summary:**

This paper tackles an important and common challenge of boundary ambiguity in the video action localization task. The authors adopt the encoder-decoder framework as DETR for embedding video/caption features and predicting the boundary locations. The proposed denoiseloc aims at regress more precise boundaries by noise injection. Extensive experiments to demonstrate the effectiveness of denoiseloc.

**Strengths:**

+ The inspiration of boundary-denoising training approach has good novelty.
+ This paper is well-organized and the proposed method achieves good performance.

**Weaknesses:**

- This paper uses a complex symbol system, which makes it difficult to read. \epsilon presents the number of fixed consecutive frames and then it presents a vector of a span. n, which represents a quantitative index, is sometimes a subscript and sometimes a superscript.
- The process of denoising is unclear. Which loss function is used for boundary denoising? The core technology is a proposal augmentation strategy to obtain more candidate proposals for training?
- Missing related works of boundary ambiguity and temporal grounding.

Wang Z, Gao Z, Wang L, et al. Boundary-aware cascade networks for temporal action segmentation[C]. ECCV2020.

Xia K, Wang L, Zhou S, et al. Learning to refactor action and co-occurrence features for temporal action localization[C]. CVPR2022.
- Typo. L5 of Sec. 3.2.

**Questions:**

- What is the definition of an action span or temporal span?
- What do 0.25 and 0.75 mean in the Sec. 3.2.3? Negative proposal set is from the inside or outside of the ground truth?

---

> ### Author Response · Authors · 2023-11-21
>
> **General Comment**
>
> We thank the reviewer for his time dedicated to assessing our work. We are pleased the reviewer acknowledged the soundness of our proposed solution and our presentation. Moreover, the reviewer's belief that our work makes a valuable contribution to the community is especially gratifying.
>
> In response to the concerns raised, we have incorporated the relevant citations in Sections 1 and 2, as suggested by the reviewer. Furthermore, we have addressed the mentioned typo and made a few additional corrections. We have also refined the notation in Section 3 of the manuscript to enhance clarity. We assure the reviewer that the adjustments are minor and do not impact any of the equations. We kindly invite the reviewer to review these modifications, which encompass all the suggestions and hope that they will find these changes to be satisfactory. Note that all our changes are highlighted in blue in the revised manuscript for the convenience of the reviewers.
>
> Additionally, we would like to clarify the boundary-related losses adopted in this work, and presented in Section 3.3, are L1 loss and gIoU loss. These losses directly affect and guide the denoising process.
>
> Finally, our approach can be perceived as an advanced form of proposal augmentation. However, it is important to emphasize that the core concept involves controlling the noise level and adjusting the ground truth accordingly while training the model to predict the original temporal span under these specific conditions.
>
> **Question 1: Temporal span definition.**
>
> We are pleased to inform the reviewer we added a short sentence in the problem formulation to formalize the definition of temporal span. We have also redacted the document to only use the naming “temporal span,” removing any instance of “action span.” We are sure this improved clarity. We thank the reviewer for the suggestion.
>
> **Question 2: Clarification from Sec. 3.2.3.**
>
> When creating negative temporal spans, for simplicity, we formulate the set as adding noise to a fixed proposal with temporal span (0.5, 0.5) in center-width coordinates. These coordinates translate to (0.25, 0.75) in (t_start, t_end) convention. We find this simple strategy to be effective and robust to different levels of noise.

---

> > ### Comment · Reviewer_uvci · 2023-11-23
> > **Post-rebuttal Comment**
> >
> > Thanks to the authors for their meticulous explanation and clarification. The author's modification helps the reader to understand the method of this paper better. Finally, l raise my score to 6.

---

### Official Review · Reviewer_UMQX · 2023-11-01

**Soundness:** 3 good
**Presentation:** 4 excellent
**Contribution:** 2 fair
**Rating:** 6
**Confidence:** 5

**Summary:**

This paper tackles the problem of video activity localization, specifically given language descriptions. The main challenge of this task is boundary ambiguity caused by the annotator subjectivity and the smoothness of temporal events. To this end, the authors design a novel framework, named denoiseloc, aiming to progressively refine the moment predictions. To facilitate the model training, boundary-denoising training scheme is adopted, which encourages the decoder to reconstruct ground truths from the noisy moments. In the experiments on two benchmarks, MAD and QVHighlights, the effectiveness of the proposed method is validated.

**Strengths:**

+ The paper is well-written and easy to follow with good readability.
+ The figures well represent the proposed method, helping the understanding.
+ The proposed approach surpasses the strong competitors on both benchmarks.
+ The comparison between Denoiseloc and Diffuseloc is interesting, and brings valuable insights.

**Weaknesses:**

- Some important details of the method are missing. In its current form, the information about the model is insufficient in the manuscript.

(1) DETR-like approaches conventionally adopt the moment representation of (center, width). In contrast, the authors stated that they express a moment as start and end. In this case, the start position can be predicted as a larger value than the end position. How do the authors handle this?

(2) The details of temporal ROI alignment and DynamicConv layer are missing. I would like to suggest the authors to include graphical illustrations of these operations at least in the appendix for help the understanding.

(3) In the boundary-denoising process, the model generates two types of noise-injected proposals, i.e., positive and negative sets. To my understanding, the proposals in the positive set have their corresponding ground truths by design, so the model learns to recover the ground-truth moments from them. However, there is a lack of explanations about the role of the proposals in the negative set. Are they also used to recover ground truths? Or do they serve as negative samples for classification? If the former is the case, how is the matching performed? In addition, what happens if they overlap with ground truths? Will it disturb the training?

- The comparisons with existing DETR-approaches seem not fair. To my knowledge, the DETR-based approaches (e.g., Moment-DETR and UMT) leverage four encoder layers and two decoder layers with a total of 10 queries on QVHighlights. On the other hand, the proposed architecture utilizes (at most) four times more encoder/decoder layers than those of the competitors, and three times more moment queries than those of the competitors. This makes it unclear whether the performance gains come from increased parameters or the proposed algorithm, and it is highly encouraged to perform comparisons under the same setting. In addition, comparisons on the computational cost and the memory consumption will be beneficial. Meanwhile, one of the state-of-the-art method, QD-DETR [1], is missing in the comparison table. If included, the proposed method shows inferior performances even with more layers and more queries.

[1] Moon et al., “Query-Dependent Video Representation for Moment Retrieval and Highlight Detection”, CVPR, 2023.

**Questions:**

Please refer to the Weakness section.

---

> ### Author Response · Authors · 2023-11-21
>
> **General Comment**
>
> We sincerely appreciate the reviewer's thorough assessment of our manuscript and are happy to hear that they found our proposed method both sound and our presentation of high quality. It is positive to know that our experimental setup effectively demonstrates the superiority of our method over current state-of-the-art approaches and that no further essential ablation studies are required. Moreover, the reviewer's belief that our work makes a valuable contribution to the community is especially gratifying.
>
> Below, we have addressed each of the concerns raised. We hope these responses provide comprehensive clarifications. Should there be any further queries or need for additional clarification, we warmly invite the reviewer to reach out. Note, that all our changes are highlighted in blue in the revised manuscript for the convenience of the reviewers.
>
> **1 - Temporal span representation.**
>
> The reviewer is concerned about a possible invalid temporal span due to noise injection. First, we would like to clarify that our manuscript uses the start-end convention (t_start, t_end) for temporal span representation, aligning with standard practices in the field. This choice was made for consistency with the literature on this task.
>
> In our implementation, these temporal spans are initially expressed as (t_start, t_end) and then converted into a center-width (center, width) format before adding the noise.  To prevent the generation of empty/invalid temporal spans, the width is clamped to a minimum of 1e-4 after adding noise. Furthermore, our implementation of Region of Interest (ROI) alignment can ensure that only valid features are computed to extract proposal features, even in cases where the span width exceeds the video duration.
>
>
> For a more detailed understanding, we invite the reviewer to consult the code provided in the supplementary material if deemed necessary. Particular attention may be directed to the functions span_cxw_to_xx() and span_cxw_to_xx() utilized during the forward pass operations, as illustrated in files code/model.py:line196, code/model.py:line214.
>
>
> It is also important to note that before the computation of the loss, the temporal coordinates are converted back to the original (t_start, t_end) format. While the loss computation could, in theory, be performed in the center-width format, we consider this a matter of implementation detail and chose the current approach for consistency and clarity.
>
>
>
> **2- ROI alignment and DynamicConv.**
>
> We would like to clarify that these two modules were not introduced in this work but were adopted from previous work. We are happy to reference further the original manuscripts that introduce them.
> DynamicConv is directly borrowed from Sparse-R-CNN [1], while ROI alignment was initially introduced in its 2D form in Mask-R-CNN [2] (Figure 3) and successively generalized for 1D signals in [3]. We have updated our Appendix in Section G to include graphical visualizations of the two modules.
>
>
> **3- Boundary-denoising process. **
>
> We thank the reviewer for the opportunity to elaborate on the role of negative samples in our boundary-denoising process and have taken this chance to enrich the manuscript with additional details in Section 3.2.3.
>
> The distinction between positive and negative samples is only relevant when generating noisy temporal spans. For positive samples, noise is added to the spans annotated in the dataset. Conversely, for negative samples, noise is introduced to predetermined temporal spans. Following this, both sets are combined and processed through the transformer decoder, as detailed in Section 3.2.3.
>
> This unified set then undergoes our denoising process before being presented to the Hungarian Matcher for loss computation. Notably, during this stage, negative samples have little probability of being paired with growth truths. Such pairings, when they occur, are effectively penalized by our chosen loss functions.
>
> The reviewer cleverly noted the possibility of negative samples being denoised in a manner that causes them to overlap with positive ones. While this is a possibility, we would like to emphasize that, in practice, we have not observed this phenomenon to disrupt the training process.
>
> **4- Comparison Fairness.**
>
> We have structured our response to the reviewer's concern regarding comparison fairness into distinct parts.

---

> > ### Author Response · Authors · 2023-11-21
> > **Comparison Fairness and Reference**
> >
> > **4- Comparison Fairness.**
> >
> > We structure our response to the reviewer's concern regarding comparison fairness here.
> >
> > *4.1-Learnable query number*
> >
> > We acknowledge the reviewer's observation that Moment-DETR employs fewer queries. This is corroborated by findings in Table 8 of [4], where it is noted that an increase in the number of queries actually degrades performance in their setup. In contrast, our experiments, as shown in Table 3 of our main paper, indicate a positive correlation between the number of queries and the performance of our system.
> >
> > Given these findings, we argue that our comparison is fair as we select the best regime for both settings. Applying fewer queries to our system, DenoiseLoc would place it at a comparable disadvantage, as would evaluating Moment-DETR with an increased number of queries. It is also pertinent to note that integrating multiple queries into our system does not impose significant computational burdens due to the efficiency of parallel computation.
> >
> > *4.2-Model size and computation cost*
> >
> > We compare the running time and peak GPU memory of our method and baseline (Moment-DETR) in the table below. The peak memory is the max value reported from the NVIDIA system management interface (i.e., nvidia-smi). The pipeline running time is from the video beginning to load to the localization being predicted, averaging over 100 runs. The localization running time is measured only from the video feature being fed into the localization model until the model gives the output, averaging over 1000 runs. In the last column, we also report the model performance, average mAP, on the val-split of QV-Highlights.
> >
> > | Method                      | Peak GPU memory (MB) | pipeline runtime (ms) | localization runtime (ms) | Performance (AmAP) |
> > |-----------------------------|----------------------|-----------------------|---------------------------|--------------------|
> > | baseline (Moment-DETR)      | 1843                 | 984.84                | 5.22                      | 32.20              |
> > | Ours (w. 8 decoder layers) | 2145                 | 1057.96               | 19.47                     | 45.29              |
> > | Ours (w. 1 decoder layers) | 1851                 | 936.72                | 5.24                      | 38.48              |
> >
> > The peak GPU memory usage of our model (8 layers) increased from 1843 MB to 2145 MB. The run time of our localization model (8 layers) runs a bit slower than the baseline (19.47 ms vs 5.22 ms)
> > However,
> > (1) Our model in 1 decoder layer setting takes similar running time and GPU memory to the baseline but reaches higher performance.
> > (2) Compared with the entire video localization pipeline, the major latency is on video frame extraction and frame/language feature extraction, while the localization process takes less than 1% of the whole latency.
> >
> > Therefore, we consider the extra computation overhead to be negligible.
> >
> > *4.3- Comparison with QD-DETR [5]*
> >
> > We thank the reviewer for pointing out this relevant related work. We included its citation and comparison in our manuscript. Kindly see Section 4.4.  For convenience, we present a summary of this comparison here.
> >
> > | Model      | Modality       | R1,IoU=0.5 | R1,IoU=0.7 | mAP,IoU=0.5 | mAP,IoU=0.75 | mAP,Avg |
> > |------------|----------------|:----------:|:----------:|:-----------:|:------------:|:-------:|
> > |   QD-DETR  |     Visual     |    62.40   |    44.98   |    62.52    |     39.88    |  39.86  |
> > |   QD-DETR  | Visual + Audio |    63.06   |    45.10   |    63.04    |     40.10    |  40.19  |
> > | DenoiseLoc |     Visual     |    59.27   |    45.07   |    61.30    |     43.07    |  42.96  |
> >
> > This comparison highlights DenoiseLoc's capability to achieve more precise boundary detection, particularly evident in the superior performance at tighter IoUs. While QD-DETR demonstrates competitive performance, it's noteworthy that it achieves comparable recall rates at an IoU of 0.7 only when incorporating an additional audio modality. In contrast, DenoiseLoc outperforms QD-DETR in terms of mAP at IoU=0.75, even when QD-DETR utilizes both audio and visual inputs. Furthermore, our average mAP exceeds that of QD-DETR by over 2.7%.
> >
> > **Reference**
> >
> > [1] Sun, Peize, et al. "Sparse r-cnn: End-to-end object detection with learnable proposals." Proceedings of the IEEE/CVF conference on computer vision and pattern recognition. 2021.
> >
> > [2] He, Kaiming, et al. "Mask r-cnn." Proceedings of the IEEE international conference on computer vision. 2017.
> >
> > [3] Xu, Mengmeng, et al. "G-tad: Sub-graph localization for temporal action detection." Proceedings of the IEEE/CVF conference on computer vision and pattern recognition. 2020.
> >
> > [4] Lei, Jie, Tamara L. Berg, and Mohit Bansal. "Detecting moments and highlights in videos via natural language queries." Advances in Neural Information Processing Systems 34 (2021): 11846-11858.
> >
> > [5] Moon et al., “Query-Dependent Video Representation for Moment Retrieval and Highlight Detection”, CVPR, 2023.

---

> ### Comment · Reviewer_UMQX · 2023-11-23
> **Post-rebuttal Comment**
>
> I thank the authors for providing the careful response.
>
> The added details and graphical illustrations are indeed helpful, although I am not sure whether such a large modification in the manuscript is allowed during the rebuttal.
>
> After all, most of my concerns are addressed, but I agree with some points of Reviewer 61UD on the other hand.
>
> Hence, I will raise my score to 6, yet still remaining on the borderline.

---

> > ### Author Response · Authors · 2023-11-23
> >
> > We thank the reviewer for their continuous support and feedback, allowing us to improve the clarity of our work. We polished our manuscript based on the suggestions of all reviewers, including (1) rectifying formatting issues, (2) correcting typos, (3) rephrasing some existing texts to improve the readability and clarity, (4) providing additional visualizations in appendix with correlated short descriptions for the benefit of comprehensive explanations.
> > The largest modifications are relegated to the appendix of the manuscript. Such documentation does not modify the main content of the paper, and it is only in support of a thorough presentation.
> >
> > Additionally, we carefully reviewed the ICLR24 author guidelines, where there is no specification on the number or magnitude of modifications the authors can make to the original draft based on the reviewers’ comments. The only limitations imposed by the organization committee pertain to the title and the abstract, which we did not modify.
> >
> > Finally, we agree with 61UD that our discovery is counter-intuitive, as diffusion-based work is becoming a new trend in multiple areas. However, our experiments show that with a good model architecture, single-step denoising can be sufficient. Our intuition is that the temporal localization output space (defined by two real values) is much simpler than an image-generation output space (224x224x3 values). Therefore, single-step denoising can be enough for the task solution, and no benefit is brought by applying multiple-denoising steps. We look forward to seeing the community further investigate this approach based on our findings.

---

### Meta-Review · Area_Chair_Un7H · 2023-12-15

**Metareview:**

reviewer prefers to raise the score and thinks the score should be 3-5, and finally maintain the initial score. Based on the negative comments, the major issues are: (1) Some details are unclear, such as the clarity of the proposed boundary-denoising training method (Sec 3.2.3). (2) Please improve the paper writing in the final version. After reading the comments, the AC recommends to accept this paper and encourages the authors to take the comments into consideration in their final version.

**Justification For Why Not Higher Score:**

Please see the detailed comments.

**Justification For Why Not Lower Score:**

Please see the detailed comments.

---

### Decision · Program_Chairs · 2024-01-16

Accept (poster)